# Mobile Robots Exploration via Deep Reinforcement Learning

**MA Han**
Department of Electronic Engineering
The Chinese University of Hong Kong
Shatin, Hong Kong
hanma@link.cuhk.edu.hk

**XU Peng**
Department of Electronic Engineering
The Chinese University of Hong Kong
Shatin, Hong Kong
1155136367@link.cuhk.edu.hk

## Abstract

In this paper, we try to solve the mobile robot exploration problem in a 2D indoor office environment by deep reinforcement learning. Finally, we find that PPO with Convolutional Neural Network (CNN) and Dense layers converges well in our problem. The video link of our current work is here.

## 1 Introduction

Mapping and exploration of a priori unknown environment is a crucial capability for mobile robot autonomy. Recent years, information-theoretic exploration methods have been developed to reason about the information gain when robot takes particular action during exploration. In this course project, we intend to guide robot autonomous exploration by reinforcement learning. Leveraging the advantages of the reinforcement learning, we expect that the complex mutual information entropy computation can be bypassed and the mobile robot can explore a whole indoor environment autonomously and fast without collision.

### 1.1 Related Work

Recent years, a lot of frontier-based mobile robot exploration strategies have been proposed. Tang et al.[1] propose a exploration strategy based on the wavefront algorithm which is used to find the closest frontier point in short time. The mobile robot moves to the frontier point along the path planned by the wavefront algorithm, once the next frontier point is determined. Bai et al. [2] utilize previous trained deep neural networks to predict the optimal or near optimal informative sensing action. The intensive ray casting required to compute mutual information can be avoided by their method. However, in supervised learning, the heavy data labelling work is inevitable and the states not covered by the training data may be not recognized by the robot. Thus, researchers introduce reinforcement learning into the robot exploration tasks.

Since the traditional RL techniques suffer from the curse of dimensionality due to large state space or large action space and the mobile robot exploration problems always have large state space, the DRL is widely used in mobile robot exploration problem. In [3], Niroui et al. combine deep reinforcement learning with the traditional frontier-based exploration approach and the experiments show that their method can effectively determine the appropriate frontier locations. In [4], method based on the Deep Q-Network framework takes only the raw depth image as input to estimate the Q values corresponding to all actions. Zhang et al. [5] design a DRL network which uses raw sensory data from the robot's onboard sensors to determine a series of primitive navigation actions for the robot to execute. Botteghi et al. [6] also propose a DRL based method for robot navigation, which learn continuous velocity commands from raw sensory data.

The learning-based methods endow the robot with the cognitive ability. Among them, the supervised learning improves this ability via the labelled data, while the reinforcement learning improves by the

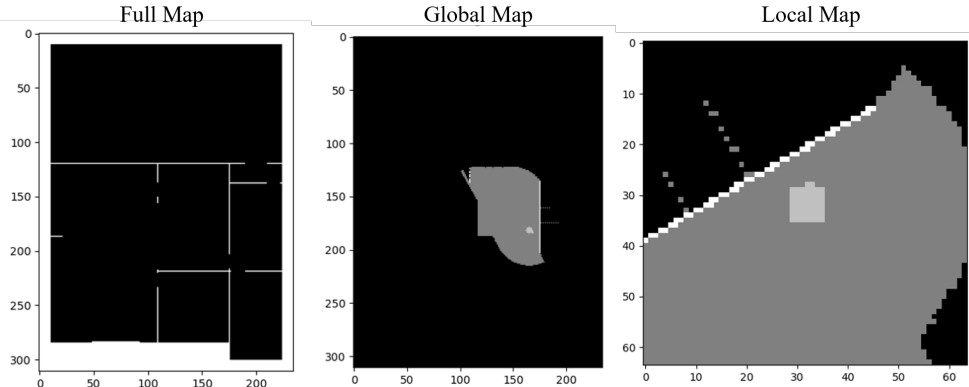

Figure 1: The figures from left to right are examples of full map, global map and local map respectively.

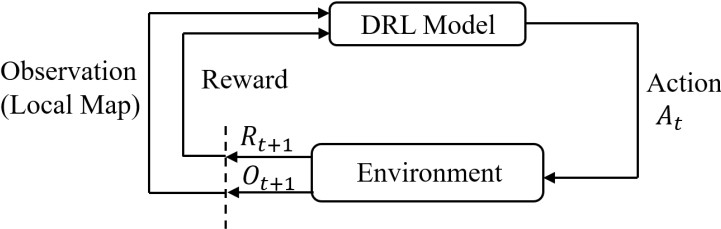

Figure 2: DRL framework

feedback from the environment. Inspired by aforementioned methods, we would like to solve the mobile robot autonomous exploration problem by DRL methods. The rest of this report is organized as follows. Section 2 formulate the mobile robot autonomous exploration problem. In section 3, we describe the experiment setup in detail and discuss the current progress.

## 2   Problem Formulation

In order to focus more on the implementation of DRL algorithm, we use a well build environment called HouseExpo [7] which will be introduced detailedly in section 3. As Figure 1 shows, the full map is one of the indoor environment and the white lines in the figure represent walls of the environment. Uncertain area and free space are indicated in black and gray, respectively. At the middle of Figure 1, the global map shows a partial explored map of the full map. Local map in Figure 1 is the observation of the environment and its center is decided by the agent, i.e. the position and direction of the agent is fixed in the local map. The size of the local map is also fixed which is $64 \times 64$ pixels. As the exploration process goes on, the gray area will become larger and larger. We assume that the mobile robot is mounted with a range sensor and the gray area means it has been measured by the range sensor. The exploration process continues until the gray area accounts for more than 95% of the full map.

The framework of the model-free DRL can be summarised as Figure 2. In our problem, the observation is the local map and the actions are turn left, turn right, and go forward. The reason for taking the local map as the observation is that the indoor environments may have different scales but they have similar local features. Thus, we hope that the robot can take actions according to the local map patterns. For the definition of the reward, if the robot collides with obstacles, the reward is -1. Otherwise, to encourage the robot to explore unknown area, the reward is the weighted sum of observation reward $r_o$ and action reward $r_a$. The observation reward is related to the newly discovered area and action reward is related to the forward action. Referring to [7], the reward function is defined as

$$r_t = \begin{cases} -1.0, & if\ a\ collision\ happens\ at\ t, \\ \alpha_o r_o + \alpha_a r_a, & if\ no\ collision\ happens\ at\ t. \end{cases} \tag{1}$$

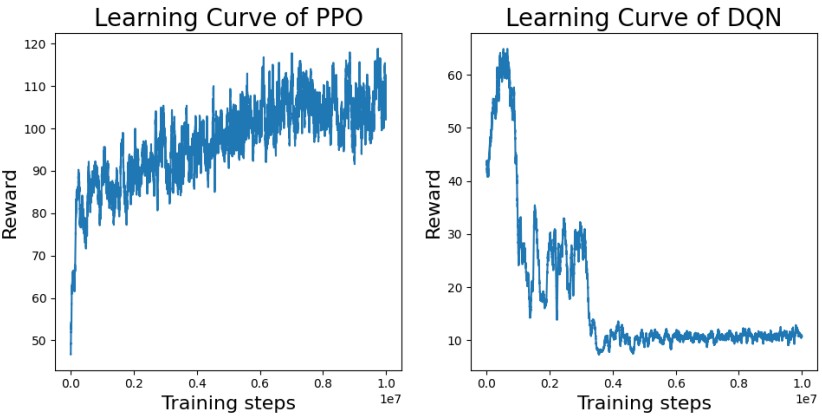

Figure 3: The learning curves of deep reinforcement learning methods at training stage

where $\alpha_o = 0.9$ and $\alpha_a = 0.1$.

## 3 Experiment Setup and Result

Li et al. [7] develops a lightweight and efficient simulation platform PseudoSLAM to accelerate the data generation process, thereby speeding up the training process. The dataset HouseExpo [7] is also built, which is a large-scale 2D floor plan dataset consisting of a large mount of human-designed 2D house blueprints, ranging from single-room studios to multi-room houses. PseudoSLAM is a simulator with OpenAI Gym-compatible [8] interface which simulates SLAM and the navigation process in an unknown 2D environment. It reads the data from HouseExpo, creates the corresponding 2D environment and generates a mobile robot to carry on navigation and exploration tasks in the indoor environment. The parameters of the simulator are specified in the configuration file, including the number and size of obstacles, robot linear movement and angular movement in each step, sensor configuration and map resolution, and state sizes. We can change these parameters for different applications.

PseudoSLAM can achieve competitive mapping result as SLAM with much less time cost. In the PseudoSLAM simulation platform, occupancy grid maps are utilized as observation instead of the typical observation from sensory data. The occupancy grid map is an output form of the SLAM process which can avoid the discrepancy problem between simulation and real world experiment. The grip map consists of three states, i.e. free space, obstacles, and uncertain areas, which are represented by different pixel values. Actions in PseudoSLAM are discrete. As mentioned in the section 2, the action space consists of three directional movements, i.e. forward, left rotation and right rotation. The pose of mobile robot is updated according to its action, and a sector $S_t$ centered at the position of the robot with a radius of the sensor range and angle of the filed of view is cropped to hide the areas behind obstacles or walls. The newly explored area $r_o$ is merged to the global map built at time step $t - 1$, which is the sum of previous explored area.

PseudoSLAM abstracts away low-level sensor data processing so that we can focus on performing high-level strategic policy based on the built maps. Using the OpenAI Gym-compatible interface provided by the simulator, we can easily integrate the-state-of-the-art reinforcement learning methods.

For the task of autonomous exploration, we let the robot explore the whole given map which is free of obstacles. The robot should accomplish the goal to explore the whole area of indoor environments with different layouts. This problem is formulated as Markov decision-making process. We train the policy utilizing some deep reinforcement learning methods, such as Proximal Policy Optimization (PPO) [9] and DQN [10], in a set of indoor environments and then test the learned policies in these environments.

The learned policies focus on reasoning about the topology of the environment. We train the robot to use spacial knowledge [8] to explore unknown area in indoor environments. One OpenAI Gym-compatible environment of robot exploration has been implemented [7]. To train a model using

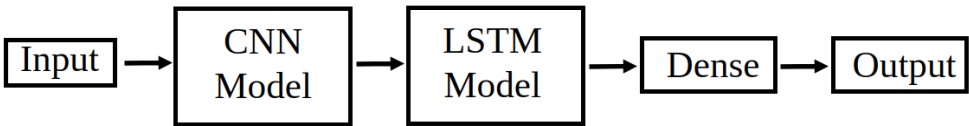

Figure 4: CNN-LSTM architecture

OpenAI baselines, we first add the environment in Gym to the register file. As Convolutional Neural Networks are widely used in computer vision domain, we use CNNs to estimate the action function and critic function in PPO and value function in DQN, respectively. PPO and DQN baselines are employed to train the neural network. In the training process, the robot observes a local rectangular map around it which contains walls. If the mobile robot reaches the maximum steps or the explored area occupies more than 95% area of the whole given global map, the robot will stop moving. The reward and the explored area are calculated.

We compare the reward generated from PPO and DQN. The learning curves are depicted in Figure 3. we can find that the PPO outperforms DQN. we also test the learned policy on the previously sampled maps.

## 4 Analysis and Current Work

In the experiment, we find that PPO can converge after many episodes, but DQN can not. PPO and DQN are both off-policy method. They first collect data into the memory using current policy and then update the policy or value functions. The differences are that PPO is critic-actor based which approximate both policy and value function, while DQN is value based which approximate only value function and then derive policy according to value function. In DQN, the agent always tries to find an optimal action, but it is hard to tell which is the best action and current best action can not guarantee the best trajectory. Thus, proximal is better in our problem. That is the intuitive why DQN is hard to converge. In figure 3, two methods use the same CNN structure as [10]. The input to the CNN consists of $4 \times 64 \times 64$ image produced by the environment. The first convolutional layer has 32 filters of $8 \times 8$ with stride 4 and applies Rectified Linear Unit (ReLU). The second convolutional layer have 64 filters of $4 \times 4$ with stride 2 and also applies Rectified Linear Unit (ReLU). The third convolutional layer has 64 filters of $3 \times 3$ with stride 1 followed by a rectified. The last hidden layer is fully-connected and consists of 512 rectifier units.

Considering that the states of the exploration process are related in time, we add a Long Short Term Memory (LSTM) with 128 units to the CNN model to capture the temporal features. The CNN-LSTM [11] [12] architecture uses CNN layers for feature extraction on input frames combined with LSTMs to support sequence prediction, which is applicable to our problem. For clearance, we draw the CNN-LSTM architecture in Figure 4. Because of the bad performance of DQN in our configuration, we use PPO to train our model. Our environment takes local map generated by laser ranger as observations, which can be considered as continuous states, whereas actions space is discrete (i.e. turn left, turn right and go forward). In PPO, the policy network outputs the logit of each action and the critic network outputs the critic value of current state. Then, the softmax of action logit values is compute, which is used to construct a categorical distribution. The action is sampled from the categorical distribution. The log probability of current action is also computed according to the categorical distribution. We also apply the clipping strategy in PPO in case that the probability ratio is too large.

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
