# OpenReview forum: "Mobile Robots Exploration via Deep Reinforcement Learning"
_CUHK.edu.hk/2021/Course/IERG5350_

### Official Review · AnonReviewer3 · 2020-12-14
**Above average paper and needs more experiments and grammar checks**

**Rating:** 6
**Confidence:** 5

**Review:**

Summary:
This paper focuses on the mobile robot exploration problem with the help of PPO and DQN. Overall, this paper is centered around the course content and well-written.

General:
1. Significance: The main contribution of this paper is to use PPO or other popular RL models to solve the classic mobile robot exploration problem. It does not contribute much to this area since I think a lot of research works are similar to the solution in the paper. But it is fine since it meets the requirements of a course project.
2. Novelty: This paper obviously lacks novelty in general since PPO is nothing new. Also, the LSTM model and Dense layer are all existing building blocks but again it is ok for a course project. In addition, this work combines the classic control problem with DRL, which is some kind of novelty from my perspective.
3. Technical quality: The paper's technical quality is above average. The reward is clearly defined. However, it fails to mention any detailed hyper-parameter setting for PPO.
4. Clarity: This paper is not that clearly written in some sense with some drawbacks though. The paper seems to be written in a rush as there is no conclusion section. Also, the current work should not be put at the end but rather in the related works section. For me, the sections are not organized in a logical way. The order needs to be rearranged.

Specific:

a. Pros:
1. CNN-LSTM architecture is reasonable to me and has some original ideas.
2. Both DQN and PPO are tried.
3. Figure 1 is very illustrative and informative.

b. Cons:
1. No GitHub codes are available.
2. Some new metrics (besides mean reward) need to be added to verify the performance of the agent.
3. More experiments are necessary.
4. Grammar errors are distributed everywhere!!

Details:
    Just list some grammar errors here: (too many just list a few here)
    Section 4, Para 1, line 4: should be 'approximates'. (two same errors in one single line!!)
    Section 4, Para 2, line 7: at the end, should be 'i.e.,' (pls check the use of 'i.e.,')

---

### Official Review · AnonReviewer2 · 2020-12-20
**Good implementation; need more experiments and comparisons.**

**Rating:** 8
**Confidence:** 3

**Review:**

Significance: This paper implements the PPO method to solve the mobile robot exploration problem in a 2D indoor office environment on a simulation platform PseudoSLAM. I think the research on this strategy is very important due to the current development of mobile robots.

Novelty: Add an LSTM model to capture the temporal features.

Technical quality: Clear defined reward function and methodology. Determine the reward function based on collision and implement the PPO and DQN in this project.

Clarity: This paper is clear to follow, but needs to rearrange some contents.

Specific:

a. Pros:

1. Use CNN-LSTM architecture to capture the temporal features.
2. The description of the architecture is very clear.

b. Cons:

1. Need a conclusion section.
2. More experiments and comparisons.

---

### Official Review · AnonReviewer1 · 2020-12-20
**This article uses PPO and CNN to solve the mobile robot exploration problem in 2D environment.**

**Rating:** 7
**Confidence:** 4

**Review:**

General:
1. Significance: This article uses PPO and CNN to solve the mobile robot exploration problem in the 2D environment.
2. Novelty: This article combines CNN-LSTM and DRL problems and could be used in practical problems. That' great.
3. Technical quality: The environment setup is detailed. But it would be better if there are some optimizations furthermore.
4. Clarity: The content is good in general. But section4 is a little confusing, maybe the result and analysis should be put into one section and a brief conclusion is needed at last.

Specific:
1. pros: a. DQN and PPO are implemented and compared. b. CNN-LSTM is used.
2. cons: a. The structure needs to be adjusted. b. Some improvements could be implemented on the basis.